# Determination of Biologically Active Compounds and Antioxidant Capacity In Vitro in Fruit of Small Cranberries (*Vaccinium oxycoccos* L.) Growing in Natural Habitats in Lithuania

**DOI:** 10.3390/antiox13091045

**Published:** 2024-08-28

**Authors:** Mindaugas Liaudanskas, Rima Šedbarė, Valdimaras Janulis

**Affiliations:** 1Department of Pharmacognosy, Faculty of Pharmacy, Lithuanian University of Health Sciences, LT-50162 Kaunas, Lithuania; rima.sedbare@lsmu.lt (R.Š.); valdimaras.janulis@lsmu.lt (V.J.); 2Institute of Pharmaceutical Technologies, Faculty of Pharmacy, Lithuanian University of Health Sciences, LT-50162 Kaunas, Lithuania; 3Department of Analytical and Toxicological Chemistry, Faculty of Pharmacy, Lithuanian University of Health Sciences, LT-50162 Kaunas, Lithuania

**Keywords:** *Vaccinium oxycoccos*, flavonols, proanthocyanidins, anthocyanins, triterpene compounds, chlorogenic acid, antioxidant capacity in vitro, natural habitats

## Abstract

The composition of flavonols, proanthocyanidins, anthocyanins, triterpene compounds, and chlorogenic acid in small cranberry fruit samples collected in natural habitats in Lithuania and variation in the antioxidant capacity of cranberry fruit extracts was determined. This study showed that in the flavonol group, hyperoside and myricetin-3-*O*-galactoside predominated in cranberry fruit samples; in the anthocyanin group, the predominant compounds were cyanidin-3-*O*-galactoside, cyanidin-3-*O*-arabinoside, peonidin-3-*O*-galactoside, and peonidin-3-*O*-arabinoside, and in the group of triterpene compounds, ursolic acid was predominant. The highest total amounts of flavonols and anthocyanins were found in the samples collected in Čepkeliai State Strict Nature Reserve (2079.44 ± 102.99 μg/g and 6993.79 ± 350.22 μg/g, respectively). Cluster analysis of the chemical composition of small cranberry fruit samples revealed trends in the accumulation of bioactive compounds in cranberry fruit. Cranberry fruit samples collected in central Lithuania had higher levels of triterpene compounds. Statistical correlation analysis showed the strongest correlation between the quantitative composition of cyanidin-3-*O*-arabinoside and peonidin-3-*O*-arabinoside and the reducing capacity of the ethanolic extracts of the cranberry fruit samples assessed in vitro by the FRAP assay (r = 0.882, *p* < 0.01 and r = 0.805, *p* < 0.01, respectively). Summarizing the results, the geographical factor affects the variation of the quantitative composition of biologically active compounds in cranberry fruit samples.

## 1. Introduction

The genus *Vaccinium* L. (*Oxycoccus* Hill) consists of about 450 species [1]. Plants of this genus grow in natural habitats in wetland areas in the forest zone of the Northern Hemisphere [2,3,4]. The representative of this genus, small cranberry (*Vaccinium oxycoccos* L.), is widespread in the forest cenopopulations of the Baltic States, Poland, Belarus, Ukraine, and Russia [5,6]. Cranberry fruits are used as food and in traditional folk medicine for the prevention and treatment of infectious urinary tract diseases [3]. Since ancient times, the inhabitants of Lithuania have been collecting the fruits of *Vaccinium oxycoccos* in the wetland areas and were using them for food, traditional dishes, beverages, and folk medicine for the treatment and prevention of urinary tract infections.

Cranberry fruit has been found to contain bioactive compounds (flavonols, proanthocyanidins, phenolic acids, anthocyanins, triterpene compounds, etc.), which are responsible for a wide range of biological effects. Flavonols and phenolic acids have antioxidant and anticancer effects. Chlorogenic acid, which predominates in the group of phenolic acids, is responsible for antidiabetic [7,8], cardiovascular function-improving [9,10], and anti-dyslipidemic [11,12] effects. The flavonol group compound quercetin influences the proliferation of colon [13], pancreatic [14], liver [15], lung [16], breast [17], and other forms of cancer cells. One of the main advantages of the anticancer activity of quercetin is its efficacy, broad range of activity, and low toxicity compared with traditional synthetic anticancer agents [17,18]. Proanthocyanidins, triterpene compounds, and quercetin identified in cranberry fruit have been shown to inhibit tumor cell viability, induce apoptosis, and inhibit proliferation [19,20]. Compounds of the anthocyanin group lower blood glucose levels [21] and improve cardiovascular function [22]. Proanthocyanidins (especially the type A trimeric proanthocyanidin complex) found in cranberry fruits inhibit the adhesion of pathogenic microorganism strains on epithelial cells of the urinary tract [23,24]. Triterpene compounds in cranberry fruit exhibit potent anti-inflammatory activity [25] and have hepatoprotective and antiviral effects [26,27]. Ursolic acid found in cranberry fruits protects lipids from oxidation and the body from the harmful effects of oxidants of various origins [28,29].

In order to present producers and purchasers with high-quality cranberry fruit with a known composition of biologically active compounds, which the Lithuanian population uses in the preparation of traditional foods and beverages and in folk medicine, it is appropriate to assess the composition of secondary metabolites—natural antioxidants of *V. oxycoccos* fruit harvested from cenopopulations of Lithuania. It is important to identify analytical markers for flavonols, phenolic acids, anthocyanins, and triterpene compound groups, which could be used for the standardization and quality assessment of cranberry fruit raw material. Detailed studies on the phytochemical composition of cranberry fruit samples collected in different habitats can help to ensure the preparation of *V. oxycoccos* fruit raw material and to rationalize the use of plant resources. Studies on the composition of phenolic compounds in cranberry fruit samples collected in natural habitats are important and relevant, as the chemical composition and chemical compound levels in medicinal plant materials collected in different habitats may vary due to climatic, geographical, genetic, or other factors [30,31].

## 2. Materials and Methods

### 2.1. Plant Material

Small cranberry (*Vaccinium oxycoccos* L.) fruit samples were collected in 25 different places from September–October of 2021. The map of *V. oxycoccos* fruit collection sites is given in Figure 1, and the description of the sites is provided in Table 1.

The habitats where cranberry fruit samples were collected are indicated by numbers in the figures.

*V. oxycoccos* fruit samples were transported at ambient temperature (up to 3 h), then frozen at −20 °C temperature and stored until lyophilization. Cranberry fruits were lyophilized, according to Gudžinskaitė et al. [32] Loss on drying was determined according to the *European Pharmacopoeia* [33]. All presented data were recalculated for dry weight.

### 2.2. Extraction Procedure

The extraction of anthocyanins, proanthocyanidins, and flavonols was performed according to Urbstaite et al. [34,35] The extraction of triterpenic compounds was performed according to Sedbare et al. [36]

### 2.3. Spectrophotometric Analysis

All the spectrophotometric analyses were carried out with an A UV–Vis spectrophotometer (M550, Spectronic CamSpec, Garforth, UK).

#### 2.3.1. Determination of Total Proanthocyanidin Content

The total proanthocyanidin content in the extracts of cranberry fruits was evaluated by a DMCA (4-(dimethylamino)cinnamaldehyde) assay [37]. Parameters of DMCA assay for determining total proanthocyanidin content are presented in the Appendix A.

#### 2.3.2. Determination of Antioxidant Capacity

An ABTS^•+^ assay was performed using the technique described by Re et al. [38] The FRAP assay was performed according to Benzie and Strain [39]. Parameters of methods for determining antioxidant capacity are presented in the Appendix A. 

### 2.4. Chromatographic Methods

The qualitative and quantitative analyses of phenolic and triterpenic compounds were performed using a Waters ACQUITY UPLC chromatograph (Milford, MA, USA) equipped with a diode array detector (ACQUITY UPLC PDA eλ, Milford, MA, USA). The chromatographic separation was performed on an ACE C18 analytical column (100 × 2.1 mm, 1.7 μm; An Avantor ACE, ACT, Aberdeen, UK). Extracts were filtered through membrane filters with a pore size of 0.22 μm (Carl Roth GmbH, Karlsruhe, Germany) into vials. The injection volume was 1 μL. Chromatographic separation was managed, and data were recorded and processed with the Empower (Waters, Milford, MA, USA) software (version 3). The quantitative analysis was performed using the calibration curves of the external standards. Parameters of the identified compounds are presented in the Appendix A.

The separation of anthocyanins was accomplished according to Vilkickyte et al. [40] The column was operated at a constant temperature of 30 °C. The mobile phase consisted of acetonitrile (solvent A) and 10% (*v*/*v*) formic acid in water (solvent B). The flow rate was 0.5 mL/min. The following conditions of elution were applied: 0–2 min, 95% B; 2–7 min, 91% B; 7–9 min, 88% B; 9–10 min, 75% B; 10–10.5 min, 20% B; and 10.5–11 min, 20% B. Anthocyanins were quantified at 520 nm.

The analysis of chlorogenic acid and flavonols was accomplished according to Urbstaite et al. [35] The column was stored at 30 °C temperature. The separation was implemented with a mobile phase consisting of acetonitrile (solvent A) and 0.1% (*v*/*v*) formic acid in water (solvent B). The flow rate was 0.5 mL/min. The following conditions of elution were applied: 0 min, 95% B; 0–1 min, 88% B; 1–3 min, 88% B; 3–4 min, 87% B; 4–9 min, 75% B; 9–10.5 min, 70% B; 10.5–12 min, 70% B; 12–12.5 min, 10% B; 12.5–13 min, 10% B; 13–13.5 min, 95% B; and 13.5–14.5 min, 95% B. Flavonols were quantified at 360 nm, and chlorogenic acid was quantified at 330 nm.

The analysis of triterpenic compounds was performed according to Sedbare et al. [36] The column was operated at a constant temperature of 25 °C. The mobile phase consisted of methanol (solvent A) and 0.1% (*v*/*v*) formic acid in water (solvent B). The flow rate was 0.2 mL/min. The following conditions of elution were applied: 0 min, 8% B; 0–8 min, 3% B; 8–9 min, 2% B; and 9–29.5 min, 2% B. Triterpene compounds were quantified at 205 nm.

### 2.5. Data Analysis

Statistical analyses were performed using SPSS Statistics 21 (IBM, Chicago, IL, USA) and Microsoft Excel 2016 (Microsoft, Redmond, WA, USA) software. All results were expressed as mean ± standard deviation. A single-factor analysis of variance (ANOVA) along with a post hoc Tukey test was employed for statistical analysis. Differences at *p* < 0.05 were considered to be significant. Hierarchical cluster analysis applying the farthest neighbor clustering method with Euclidean distances was performed to determine the groupings of cranberries. The coefficient of variation was calculated to determine the variation in the quantitative composition of the studied bioactive compounds between samples of *V. oxycoccos* fruit collected in different habitats.

## 3. Results and Discussion

### 3.1. Determination of the Composition of Total Proanthocyanidins, Individual Flavonols, and Chlorogenic Acid

The composition of secondary metabolites, natural antioxidants, is a key parameter in assessing the quality of medicinal plant raw materials. Phenolic compounds have strong antioxidant [41,42], anticancer [43,44], antimicrobial [45,46], anti-inflammatory [47,48], and other biological effects. In order to provide high-quality plant raw material with a known composition of bioactive compounds, it is relevant to determine the composition of phenolic compounds in samples of *V. oxycoccos* fruit collected in different habitats.

Proanthocyanidins determine the effects of cranberries in the treatment of urinary tract diseases and are important for their prevention [49,50]. The scientific literature indicates that proanthocyanidins in cranberry fruit effectively inhibit the adhesion of *Escherichia coli* bacteria and other pathogenic microorganism strains on urinary tract epithelial cells [24,51]. For this reason, it is important to determine the variation in the total proanthocyanidin content in *V. oxycoccos* fruit samples collected in the natural habitats of Lithuania.

The quantitative assessment of the proanthocyanidin composition in small cranberry fruit samples by the spectrophotometric method shows that the total amount of proanthocyanidins ranges from 1.36 mg EE/g to 3.47 mg EE/g. The mean total proanthocyanidin content in all cranberry fruit samples tested was 2.31. The highest total proanthocyanidin content was found in small cranberry fruit samples collected in Rėkyva swamp habitat in the Šiauliai district, in samples collected near the Varninkai educational trail habitat in the Trakai district, and in samples collected in the Amalva forest habitat in Marijampolė district. The results of the variation in total proanthocyanidin content in samples of small cranberry fruit collected in natural habitats are presented in Figure 2. In order to assess the variation in total proanthocyanidin content in samples of small cranberry fruit collected in different habitats, a coefficient of variation (22.40%) was calculated, which showed an average variation in total proanthocyanidin content in the tested samples.

Šedbarė et al. investigated changes in proanthocyanidin content in small cranberry fruit samples collected in protected areas of Lithuania. The estimates of the total amount of proanthocyanidins reported in their publication (0.92–3.4 mg EE/g) are almost identical to our estimates [52]. The total proanthocyanidin content of *V. macrocarpon* fruit samples ranged from 1727.40 μg EE/g to 3386.94 μg EE/g [53].

Flavonols have antioxidant [54,55], anti-inflammatory [54,56], anticancer [57,58], antibacterial [59,60], antiviral [61,62], cardioprotective [63,64], and anti-allergic [65,66] effects. Of the group of flavonol compounds found in cranberries, quercetin strongly suppresses the nuclear factor ĸB-pathway [67] and, therefore, may potentially be valuable for the prevention and treatment of rheumatoid arthritis and other inflammatory conditions. Myricetin found in small cranberry fruit has been shown to suppress skin [68], bladder [69], and pancreatic cancer cells through different mechanisms [70].

Due to the broad spectrum of biological effects of phenolic compounds, it is relevant to investigate the variation in the composition of proanthocyanidins and flavonols in small cranberry fruit collected in natural habitats in Lithuania.

In small cranberry fruit samples, we identified and quantified chlorogenic acid belonging to the phenolic acid group, the compounds of the flavonol group myricetin and its glycoside myricetin-3-*O*-galactoside, and quercetin and its glycosides hyperoside, isoquercitrin, avicularin, and quercetin-3-*O*-arabinopyranoside, the quantified composition of which is presented in Figure 3.

The total amount of flavonol compounds quantified in *V. oxycoccos* fruit samples ranges from 760.97 μg/g to 2079.44 μg/g. The mean of the total amount of flavonol compounds detected is 1194.35 μg/g. The percentage of flavonol group compounds in the total amount of the quantified phenolic compounds varies from 41.32% to 95.31%. The coefficient of variation, which reflects the range of variation in the total amount of the quantified flavonol compounds in small cranberry fruit samples, is calculated to be 28.86%. The highest total amounts of flavonols were found in small cranberry fruit samples collected in the habitat of Čepkeliai Reserve in the Varėna district and in Rėkyva swamp in the Šiauliai district.

In cranberry fruit samples collected in natural habitats, hyperoside and myricetin-3-*O*-galactoside were the predominant compounds determined in the flavonol group, while the levels of other flavonols identified were lower. The percentage of hyperoside in the total amount of the quantified flavonol compounds ranges from 32.58% to 47.84%, and the percentage of myricetin-3-*O*-galactoside, from 18.69% to 38.18%.

The results of our study are supported by those obtained in previous studies conducted by Šedbarė et al. Samples of small cranberry fruit collected in protected areas of Lithuania showed that hyperoside (38.33%) and myricetin-3-*O*-galactoside (31.38%) were the most abundant compounds in the flavonol group [52]. The highest levels of hyperoside and quercetin-3-*O*-pyranoside were detected in the samples of small cranberry fruit collected in the territory of Čepkeliai Reserve in the Varėna district, and the highest level of myricetin-3-*O*-galactoside was detected in small cranberry fruit samples collected in Rėkyva swamp in the Šiauliai district. The highest level of isoquercitrin was determined in *V. oxycoccos* fruit samples collected in the habitat of the Dubrava educational trail in the Kaunas district. The highest levels of avicularin and quercitrin were detected in small cranberry fruit samples collected in the Rėkyva swamp in the Šiauliai region. The highest levels of quercetin and myricetin were detected in small cranberry fruit samples collected in the Lake Aklasis shoreline habitat in the Jonava district.

Oszmiański et al. found similar levels of hyperoside (77.1–137.8 mg/100 g) and quercitrin (48.4–70.0 mg/100 g) in large cranberry fruit samples [71]. Another paper by Polish researchers reported isoquercitrin levels (4.8–11.5 mg/100 g) in large cranberry fruits similar to those found in our study, while myricetin-3-*O*-galactoside levels (156.5–348.4 mg/100 g) were higher than those found in our small cranberry samples [72]. Urbstaite et al. reported that the avicularin amount of American cranberry fruit samples ranged from 293.79 μg/g to 613.80 μg/g. Quercitrin content in *V. macrocarpon* fruit samples ranged from 227.49 μg/g to 413.68 μg/g and was higher than determined in small cranberry fruit samples in this research [35].

Chlorogenic acid has potent antioxidant activity [73,74] as well as neuroprotective [75,76], hepatoprotective [12,77], and nephroprotective [78,79] effects and also regulates carbohydrate and lipid metabolism [74]. For this reason, it is relevant to ascertain the composition of this phenolic acid in *V. oxycoccos* fruit samples using modern methods of chromatographic analysis.

The amount of chlorogenic acid in small cranberry fruit samples collected in natural habitats ranges from 76.09 μg/g to 1428.55 μg/g. The average of the estimates of the quantitative composition of chlorogenic acid in all the cranberry fruit samples analyzed is 761.78 μg/g. The highest amounts of chlorogenic acid were found in *V. oxycoccos* fruit samples collected in the Amalva forest in the Marijampolė district, in the Smalininkai neighborhood in the Jurbarkas district, and in the Kamanai Reserve in the Akmenė district. Šedbarė et al. found similar levels of chlorogenic acid (17–1224 μg/g) in samples of small cranberry fruit collected in protected areas of Lithuania [52]. Arvinte and Amariei reported that the amount of chlorogenic acid determined in small cranberry fruit samples collected in Rumania was 0.42 mg/g [80]. Stobnicka and Gniewosz found that chlorogenic acid determined in *V. oxyccocos* fruit samples collected in Poland was 96.3 mg/100 g [81]. Urbstaite et al. determined that chlorogenic acid amounts in American cranberry fruit samples varied from 119.14 μg/g to 472.97 μg/g [35].

The calculation of the coefficient of variation revealed a very large variation (as much as 50.66%) in the estimates of the quantitative composition of chlorogenic acid. This significant variation could be due to the peculiarities of the geographical location of cranberry habitats. The variation in the quantitative composition of chlorogenic acid in small cranberry fruit samples collected in natural habitats is presented in Figure 4.

Summarizing the studies on the variation of the quantitative composition of proanthocyanidins, flavonols, and chlorogenic acid in small cranberry fruit samples collected in natural habitats in Lithuania, it can be stated that the highest total amounts of proanthocyanidins were found in Rėkyva swamp in northern Lithuania, while the highest total amounts of flavonols were found in Čepkeliai Reserve in southern Lithuania and in Rėkyva swamp in northern Lithuania. The highest levels of chlorogenic acid were found in fruit samples of small cranberries collected in the Amalva forest in southwestern Lithuania.

### 3.2. Determination of the Quantitative Composition of Anthocyanins

Anthocyanins give berries the red color and determine the morphological characteristics of the fruit [82,83]. Anthocyanins and anthocyanidins have antioxidant [84,85], anticancer [86,87], anti-inflammatory [88,89], anti-ulcer [90,91], antimicrobial [92,93], antiviral [94,95], collagen-stabilizing, and capillary permeability-reducing [96] effects.

When assessing the quality of cranberry fruit, it is important to determine the composition of anthocyanins and anthocyanidins in samples of *V. oxycoccos* fruit grown in natural habitats in Lithuania, as well as the variation in the phytochemical composition of these compounds. In total, 12 anthocyanin compounds were identified in our studied samples of small cranberry fruit collected in natural habitats: delphinidin-3-*O*-galactoside, cyanidin-3-*O*-galactoside, cyanidin-3-*O*-glucoside, cyanidin-3-*O*-arabinoside, peonidin-3-*O*-galactoside, peonidin-3-*O*-glucoside, peonidin-3-*O*-arabinoside, malvidin-3-*O*-galactoside, malvidin-3-*O*-arabinoside, cyanidin, peonidin, and malvidin. Their quantitative composition is presented in Figure 5.

The total amount of anthocyanins identified and quantified in small cranberry fruit samples varied from 1384.85 to 6993.79 μg/g. The mean total anthocyanin content is 3051.89 μg/g. The estimates of the total anthocyanin content varied considerably, the coefficient of variation being 41.82%. The highest total anthocyanin content was found in samples of *V. oxycoccos* fruit collected in the Čepkeliai Reserve in the Varėna district (6993.79 ± 350.22 μg/g) and in the habitat near the shore of Lake Juodlė in the Kelmė district (6228.68 ± 302.37 μg/g). In our analyzed small cranberry fruit samples, among the identified and quantified anthocyanin compounds, cyanidin-3-*O*-galactoside, cyanidin-3-*O*-arabinoside, peonidin-3-*O*-galactoside, and peonidin-3-*O*-arabinoside were the predominant ones, while the amounts of other anthocyanin compounds were considerably lower. Data presented by Huopalahti et al. are consistent with our results. These researchers reported that cyanidin-3-arabinoside (23.1%), peonidin-3-galactoside (21.5%), cyanidin-3-galactoside (19.2%), and peonidin-3-arabinoside (14.1%) are the dominant compounds in the cranberry juice freshly prepared from *V. oxycoccos* fruit samples collected in Finland [97]. These compounds could be selected as analytical markers for the evaluation of the composition of the anthocyanin group of compounds in small cranberry fruit samples.

The highest levels of cyanidin-3-*O*-galactoside, delphinidin-3-*O*-galactoside, malvidin-3-*O*-galactoside, and malvidin-3-*O*-arabinoside were found in cranberry fruit samples collected in Rėkyva swamp in Šiauliai district. The highest levels of cyanidin-3-*O*-arabinoside, peonidin-3-*O*-arabinoside, cyanidin-3-*O*-glucoside, and peonidin-3-*O*-glucoside were found in small cranberry fruit samples collected in the Čepkeliai Reserve in the Varėna district. The highest peonidin-3-*O*-galactoside content was found in small cranberry fruit samples collected in the habitat near the shore of Lake Juodlė in the Kelmė district. The highest levels of cyanidin and peonidin were found in small cranberry fruit samples collected in the habitat on the marshy shore of Lake Aklasis in the Jonava district. The highest malvidin content was detected in small cranberry fruit samples collected in the Ežerėlis peatbog in the Kaunas district.

Mazur and Borowska reported that cyanidin-3-glucoside content in lyophilized *V. oxycoccos* fruit samples (55.2 mg/100 g) was higher than determined in our research [98]. Urbstaite et al. found that cyanidin-3-O-galactoside content in *V. macrocarpon* fruit samples varied from 0.29 mg/g to 1.92 mg/g, and peonidin-3-O-arabinoside content ranged from 0.47 mg/g to 1.48 mg/g [34]. Oszmiański et al. reported that cyanidin-3-O-arabinoside content in large cranberry fruit samples varied from 1548.9 mg/100 g to 1694.8 mg/100 g and peonidin-3-O-galactoside content ranged from 2762.3 mg/100 g to 3540.4 mg/100 g and was higher than determined in small cranberry fruit samples in this research [71].

The results of the analyses of anthocyanin content in small cranberry fruit samples collected in natural habitats in Lithuania showed that the quantitative composition of this group of compounds in *V. oxycoccos* fruit growing in Lithuania varied widely—from 1384.85 to 6993.79 μg/g. The highest total anthocyanin content was found in *V. oxycoccos* fruit samples collected in Čepkeliai Reserve in southern Lithuania and in the habitat near Lake Juodlė in northwestern Lithuania. The lowest total anthocyanin content was found in samples of small cranberry fruit collected in the habitat near Juodupė in northeastern Lithuania.

### 3.3. Determination of the Quantitative Composition of Triterpene Compounds

Triterpene compounds have been identified in the waxy layer of plant organs, which protects them from harmful environmental factors such as microorganisms, UV radiation, temperature fluctuations, etc. [99] Triterpene compounds have antioxidant [100,101], anticancer [102,103], and anti-inflammatory effects [104,105], they protect against metabolic disorders [106,107], protect the cardiovascular system [108,109], and lower blood glucose levels [110,111].

In our study, eight triterpene compounds were identified in small cranberry fruit samples collected in natural habitats in Lithuania: maslinic acid, corosolic acid, oleanolic acid, ursolic acid, α-amyrin, β-amyrin, β-sitosterol, and squalene. Their quantitative composition is presented in Figure 6.

The total amount of triterpene compounds identified and quantified in small cranberry fruit samples ranges from 5005.24 μg/g to 8252.40 μg/g. The highest total amount of triterpenes (8252.40 ± 405.96 μg/g) was determined in *V. oxycoccos* fruit samples collected in the habitat located near the Varninkai educational trail in the Trakai district. The triterpene composition in the fruit samples collected in this habitat is not statistically significantly different from that in *V. oxycoccos* fruit samples collected in other habitats in Lithuania (Figure 6). The mean of the total amount of the quantified triterpene compounds is 6775.58 μg/g. The calculated coefficient of variation (10.69%) shows a slight variation in the total triterpene compound content determined in *V. oxycoccos* fruit samples.

In the tested samples, ursolic acid predominated among the identified triterpene compounds, accounting for 64.66–77.13% of the total amount of triterpene compounds quantified. The highest content of ursolic acid was detected in samples collected in the habitat located near the Varninkai educational trail in the Trakai district. The triterpene composition of the fruit samples collected in this habitat was not statistically significantly different from that of *V. oxycoccos* fruit samples collected in other habitats (Figure 6). Šedbarė et al. found the content of ursolic acid in *V. oxycoccos* fruit samples to be 5222.6 ± 78.34 μg/g, which represented 64.80% of the total amount of the determined triterpene compounds, and thus the results obtained in our study are similar to those presented by the aforementioned authors [112]. Ursolic acid, as the predominant component of triterpene compounds, could be used as an analytical marker for the determination of the composition of triterpene compounds in small cranberry fruit samples.

The detected amounts of other identified triterpenic compounds in *V. oxycoccos* fruit samples were significantly lower. The highest levels of maslin, corosolic acids, and β-sitosterol were detected in fruit samples collected in the Rėkyva swamp in the Šiauliai district. The highest levels of oleanolic acid and squalene were detected in fruit samples collected near the Varninkai educational trail in the Trakai district. The highest amount of α-amyrin was detected in small cranberry fruit samples collected in the Kamanai Reserve in the Akmenė district. Meanwhile, β-amyrin was identified only in 6 small cranberry fruit samples collected in habitats in Lithuania. The highest β-amyrin content was found in small cranberry fruit samples collected in the Valkininkai neighborhood of Varėna district.

Oszmiański et al. determined that oleanolic acid levels in small cranberry fruit samples ranged from 894 μg/g to 1137 μg/g [71], which is in line with our results. Meanwhile, Xue et al. in their study found significantly higher levels of oleanolic acid (5910–31,240 μg/g) in cranberry fruit samples compared to those found in our study [77]. Sedbare et al. reported that maslinic acid content (46.5 ± 0.70 μg/g) in *V. oxycoccos* fruit samples was similar to those found in our study, while amounts of corosolic acid (99.6 ± 1.49 μg/g) and α-amyrin (57.1 ± 0.86 μg/g) were lower than those found in our small cranberry samples [36]. Ursolic acid content (1044–1759 mg/kg) found in *V. macrocarpon* cultivated in Poland fruit samples was lower than determined in Lithuanian small cranberries fruit samples collected in natural habitats [113]. Interspecific variation [114], genetic [115], edaphic [116], climatic [117], and other factors may cause this difference, but we did not evaluate them in this research.

Summarizing the results of the analysis of triterpene compounds in *V. oxycoccos* fruit samples collected in natural habitats in Lithuania, it can be stated that the highest total triterpenic compounds content was found in small cranberry fruit samples collected near Varninkai educational trail in southeastern Lithuania, yet the difference from the amounts of these compounds in *V. oxycoccos* fruit samples collected in the majority of the other habitats was not statistically significant.

### 3.4. Comparison of Compound Levels in Small Cranberry Fruit Samples Using Hierarchical Cluster Analysis

Hierarchical cluster analysis was performed for the comparison of the content of anthocyanins, flavonols, chlorogenic acid, proanthocyanidins, and triterpene compounds in small cranberry fruit samples. The studied *V. oxycoccos* fruit samples were divided into four clusters (Figure 7).

Many of the cranberry fruit samples in Cluster 1 were collected in habitats in northeastern Lithuania. Cluster 1 small cranberry fruit samples had statistically significantly lower anthocyanin levels compared with cranberry fruit samples of other clusters. The mean amount of flavonols in cranberry fruit samples of this cluster was also found to be lower than that in cranberry fruit samples in other clusters.

Most of the small cranberry fruit samples in Clusters 2 and 3 were collected in habitats in southern and central Lithuania. The mean amounts of anthocyanins, flavonols, and chlorogenic acid in small cranberry fruit samples of these clusters did not differ statistically significantly. The mean amounts of proanthocyanidins and triterpene compounds were statistically more significant in Cluster 3 small cranberry fruit samples compared with those detected in Cluster 2 cranberry fruit samples.

Cluster 4 consisted of small cranberry fruit samples collected in two habitats: near Lake Juodlė in the Kelmė district and Čepkeliai Reserve in the Varėna district. The small cranberry fruit samples collected in these sites were found to contain two to three times higher total levels of anthocyanins compared with those determined in fruit samples of the other clusters.

The following trends were identified based on the data from the cluster analysis: the amounts of anthocyanins and flavonols in small cranberry fruit samples collected in northeastern Lithuania were lower than those found in cranberry fruit samples collected in southern and central Lithuania; the amounts of triterpene compounds were higher in *V. oxycoccos* fruit samples collected in central Lithuania; the average amount of chlorogenic acid did not differ statistically significantly between the clusters, which indicates that the variation in the amount of chlorogenic acid in the fruit samples did not depend on the geographical location of the plant.

### 3.5. Determination of Antioxidant Capacity of Vaccinium oxycoccos Fruit Extracts

Medicinal plant materials, which are a source of natural antioxidants, and their preparations affect the human body, which is supported by scientific studies [118,119,120]. A link has been established between the consumption of antioxidant-rich plant materials and their preparations and the incidence of oncological [121,122], cardiovascular [123,124], and neurodegenerative [125,126] diseases. It is thus important to search for new raw materials that accumulate antioxidants and to carry out studies on the antioxidant effects of their extracts.

We determined the antiradical capacity of ethanolic extracts of *V. oxycoccos* fruit samples in vitro by applying the ABTS technique. This capacity ranged from 37.38 μmol TE/g to 141.28 μmol TE/g. The mean antiradical capacity of the extracts in vitro was 103.31 μmol TE/g. Samples collected in the Amalva forest in the Marijampolė district showed the strongest antiradical capacity in vitro. There was no statistically significant difference in antiradical capacity between the extracts of small cranberry fruit collected in the Rėkyva swamp in the Šiauliai district, the Valkininkai neighborhood in the Varėna district, the Smalininkai neighborhood in the Jurbarkas district, the lakeside habitat of Lake Juodlė in the Kelmė district, the Čepkeliai Reserve in the Varėna district, or the Labanoras forest in the Molėtai district (Figure 8). The coefficient of variation of the antiradical capacity of ethanolic cranberry fruit extracts in vitro was calculated to be 18.86%. Gudžinskaitė et al. found that the antiradical capacity in vitro of *V. macrocarpon* fruit samples ethanolic extracts (170.68 μmol TE/g–193.63 μmol TE/g) evaluated by the ABTS assay was stronger than determined in our research [32].

We used the FRAP method to determine the reducing capacity of ethanolic extracts of cranberry fruit samples in vitro and found that this capacity ranged from 58.37 μmol TE/g to 228.24 μmol TE/g. The mean reducing capacity of the extracts of the samples in vitro was 141.93 μmol TE/g. The strongest reducing capacity was found in ethanolic extracts of small cranberry fruit samples collected in the Čepkeliai Reserve in the Varėna district. The coefficient of variation of the reducing capacity of ethanolic cranberry fruit extracts in vitro was calculated to be 22.21%. Urbstaite et al. found that reducing capacity in vitro determined by FRAP assay of large cranberry fruit ethanolic extracts ranged from 215.23 μmol TE/g to 528.05 μmol TE/g [34]. Gudžinskaitė et al. found that reducing capacity in vitro of *V. macrocarpon* fruit samples ethanolic extracts (21.80 μmol TE/g–41.88 μmol TE/g) evaluated by FRAP spectrophotometric assay was weaker than determined in this research [32]. The results of the analysis of the antiradical and reducing capacity of ethanolic extracts of cranberry fruit samples in vitro are presented in Figure 8.

Statistical correlation analysis was performed to assess the relationship between the quantitative composition of anthocyanins, flavonols, proanthocyanidins, chlorogenic acid, and triterpene compounds in *V. oxycoccos* fruit samples and the antiradical and reducing capacity of cranberry fruit extracts in vitro.

In terms of correlation between anthocyanin compounds, the strongest positive correlation was found between the quantitative composition of cyanidin-3-*O*-arabinoside and peonidin-3-*O*-arabinoside and the reducing capacity of ethanolic extracts of cranberry fruit samples in vitro evaluated by the FRAP assay (r = 0.882, *p* < 0.01 and r = 0.805, *p* < 0.01, respectively). The ABTS assay of the antiradical capacity of anthocyanin compounds in ethanolic extracts of small cranberry fruit in vitro revealed the strongest correlation with the levels of cyanidin-3-*O*-galactoside and cyanidin-3-*O*-arabinoside (r = 0.637, *p* < 0.01 and r = 0.640, *p* < 0.01, respectively). The total amount of anthocyanins detected and the reducing capacity of the extracts of *V. oxycoccos* samples in vitro evaluated by the FRAP method correlated moderately strongly (r = 0.581, *p* < 0.01). There was a positive weak correlation (r = 0.306, *p* < 0.05) between the total anthocyanin content and the antiradical capacity of the extracts in vitro evaluated by the ABTS assay. Oszmiański et al. investigated extracts of large cranberry fruits and reported similar correlation trends: anthocyanin content moderately strongly positively correlated with the antioxidant capacity of *V. oxycoccos* fruit sample extracts in vitro evaluated by the ABTS and FRAP methods (r = 0.675 and 0.614, respectively) [113].

The correlation analysis of the reducing capacity of flavonol group compounds in ethanolic extracts of small cranberry fruit samples in vitro was evaluated using the FRAP method and showed that the strongest positive correlation was found with the quantitative composition of hyperoside and quercetin-3-*O*-arabinpyranoside (r = 0.698, *p* < 0.01 and r = 0.721, *p* < 0.01, respectively). The evaluation of the antiradical capacity of ethanolic extracts of *V. oxycoccos* fruit in vitro using the ABTS assay showed that the strongest correlation was with the levels of myricetin-3-*O*-galactoside and hyperoside (r = 0.660, *p* < 0.01 and r = 0.606, *p* < 0.01, respectively). The correlation coefficients of the antioxidant capacity of cranberry fruit extracts in vitro evaluated by the FRAP and the ABTS methods with a total amount of flavonol compounds detected in the fruit samples were 0.698, *p* < 0.01 and 0.692, *p* < 0.01, respectively.

Oszmiański et al. reported that the flavonol content of *V. macrocarpon* fruit samples strongly positively correlated (r = 0.728, *p* < 0.05) with the reducing capacity of cranberry fruit sample extracts in vitro evaluated by the FRAP assay and moderately positively correlated (r = 0.646, *p* < 0.05) with the antiradical capacity of the extracts in vitro evaluated by the ABTS assay [113].

A positive correlation between the quantitative composition of phenolic compounds in the organs of *Vaccinium* genus plants and the antioxidant capacity of their extracts was also reported by Kalın et al. [127], Seeram et al. [128], and Zheng and Wang [129]. No statistically significant correlations were found between the content of chlorogenic acid determined in *V. oxycoccos* fruit samples and the antiradical or reducing capacity of their extracts in vitro.

## 4. Conclusions

The results of the study provide new insights into the variation in the quantitative composition of secondary metabolites—natural antioxidants in *V. oxycoccos* fruit samples collected in different natural habitats of Lithuania. To summarize the results of the analyses, it can be stated that cranberry fruit samples collected in the habitat of Čepkeliai Reserve at the time of ripening showed the highest levels of anthocyanins, flavonols, and triterpene compounds. The levels of anthocyanins and flavonols found in small cranberry fruit samples collected in northeastern Lithuania were lower than those found in cranberry fruit samples collected in southern and central Lithuania. Small cranberry fruit samples collected in central Lithuania showed higher levels of triterpene compounds. However, the influence of edaphic, climatic, genetic, and other factors was not evaluated during this study.

The analyses showed that hyperoside and myricetin-3-*O*-galactoside were the predominant compounds in the flavonol group, cyanidin-3-*O*-galactoside, cyanidin-3-*O*-arabinoside, peonidin-3-*O*-galactoside, and peonidin-3-*O*-arabinoside were the predominant compounds in the anthocyanin group, while ursolic acid was found to be predominant in the group of triterpene compounds. The predominant bioactive compounds could be used as analytical markers for the quantitative assessment of the composition of small cranberry fruit samples and products derived from them by applying the HPLC method.

## Figures and Tables

**Figure 1 antioxidants-13-01045-f001:**
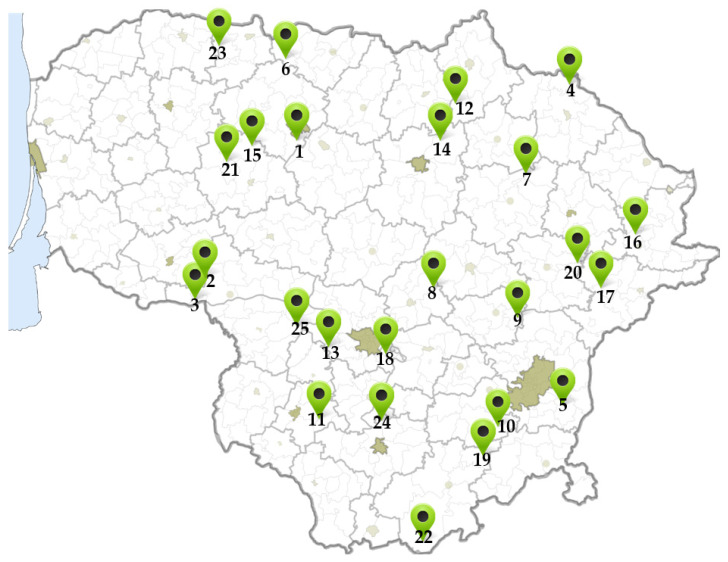
The map of small cranberry fruit collection sites.

**Figure 2 antioxidants-13-01045-f002:**
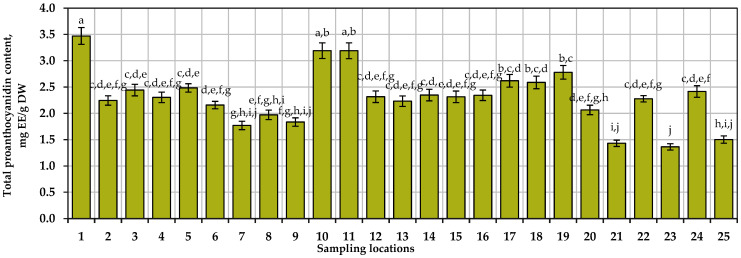
Total proanthocyanidin content in small cranberry fruit samples collected in natural habitats. Different letters (a–j) indicate statistically significant differences in total proanthocyanidin content between the tested small cranberry fruit samples (*p* < 0.05).

**Figure 3 antioxidants-13-01045-f003:**
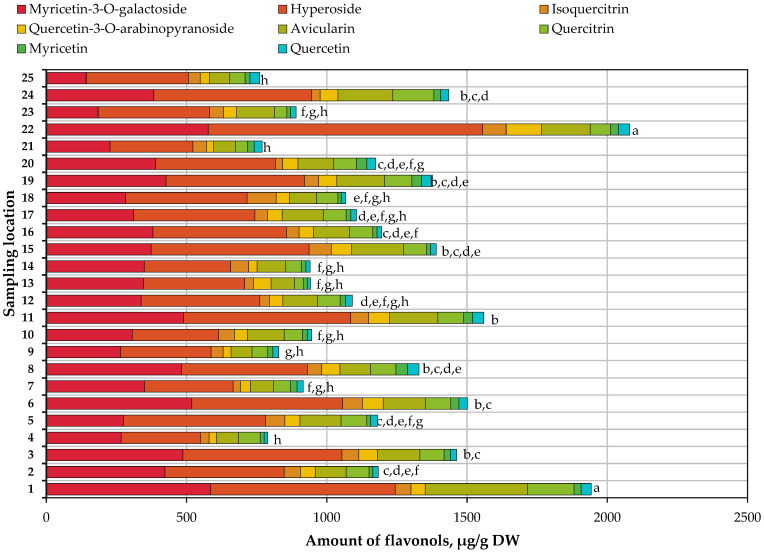
Variation in the quantitative composition of flavonol compounds in small cranberry fruit samples collected in natural habitats. The different letters indicate statistically significant differences between the values of the total flavonol content in *V. oxycoccos* fruit samples (a–h, *p* < 0.05).

**Figure 4 antioxidants-13-01045-f004:**
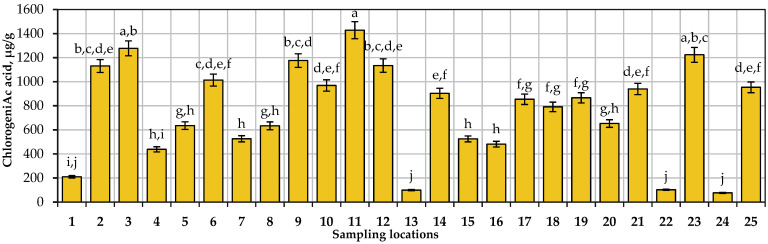
Variation in the quantitative composition of chlorogenic acid in small cranberry fruit samples collected in natural habitats. Statistically significant differences in quantitative composition of chlorogenic acid between *V. oxycoccos* fruit samples are marked with different letters (a–j, *p* < 0.05).

**Figure 5 antioxidants-13-01045-f005:**
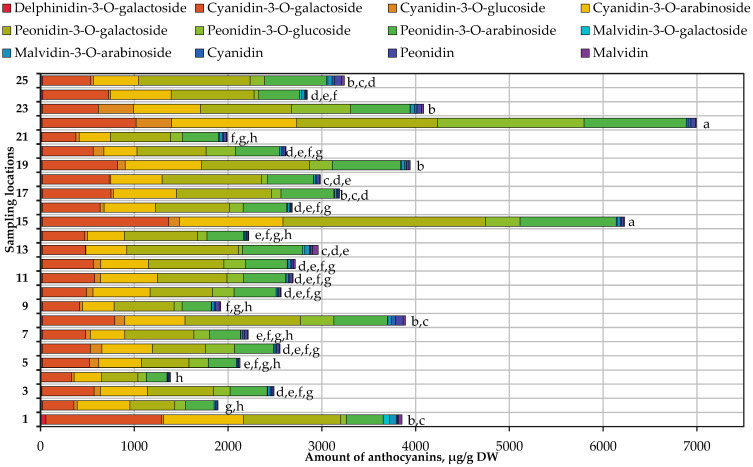
Variation in the quantitative composition of compounds of the anthocyanin group in small cranberry fruit samples collected in natural habitats. Statistically significant differences in total anthocyanin content between *V. oxycoccos* fruit samples are marked with different letters (a–h, *p* < 0.05).

**Figure 6 antioxidants-13-01045-f006:**
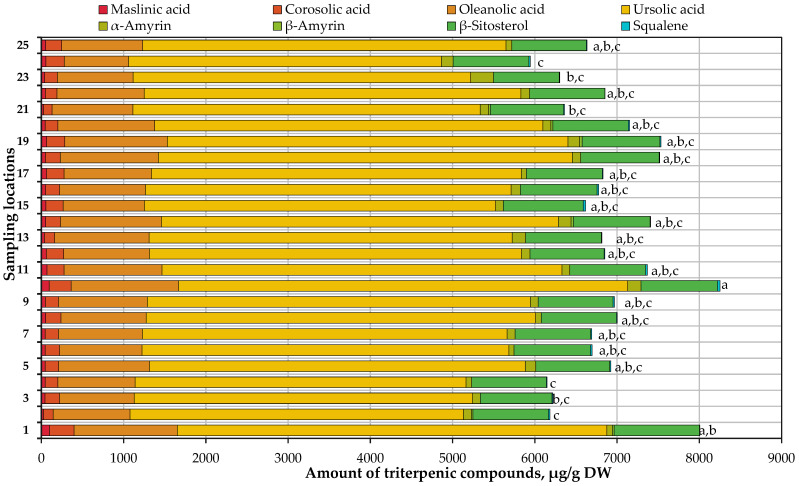
Variation in the quantitative composition of triterpene compounds in small cranberry fruit samples collected in natural habitats. The different letters indicate statistically significant differences between the values of the total amount of triterpenic compounds in *V. oxycoccos* fruit samples (a–c, *p* < 0.05).

**Figure 7 antioxidants-13-01045-f007:**
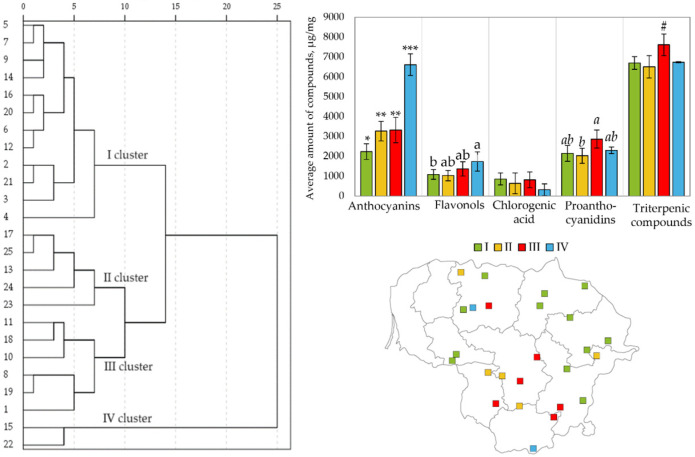
Hierarchical cluster analysis of small cranberry samples collected in natural habitats: similarity dendrogram based on the amounts of phenolic and triterpene compounds, average compound content in clusters, and distribution of cranberry fruit samples assigned to different clusters in the territory of Lithuania. Different symbols (*/**/*** for anthocyanins, a,b for flavonols, *a*,*b* for proanthocyanidins, # for triterpenic compounds) indicate statistically significant differences in the mean content of a group of compounds determined in cranberry fruit samples (*p* < 0.05).

**Figure 8 antioxidants-13-01045-f008:**
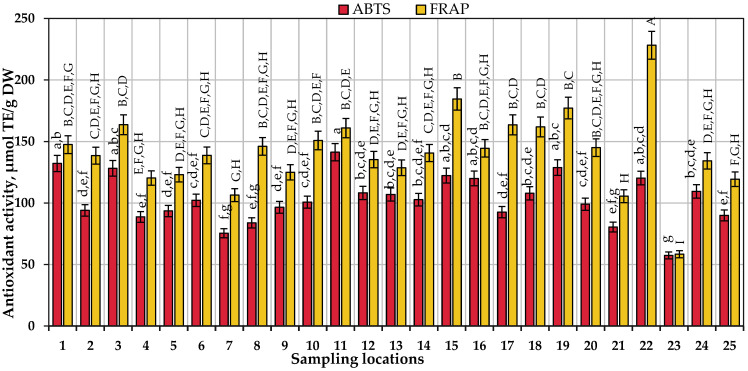
Variation in the in vitro antioxidant capacity of ethanolic extracts of small cranberry fruit samples collected in natural habitats; different letters (a–g for ABTS and A–I for FRAP) indicate statistically significant differences between the antioxidant capacity estimates of the tested cranberry fruit sample extracts (*p* < 0.05).

**Table 1 antioxidants-13-01045-t001:** Description of *V. oxycoccos* fruit harvesting areas.

No.	Habitat	Date of Harvesting
1.	Rėkyva swamp, Šiauliai district	13 September 2021
2.	Laukesa swamp, Tauragė district	13 September 2021
3.	Smalininkai, Jurbarkas district	13 September 2021
4.	Near Juodupė, Rokiškis district	13 September 2021
5.	Maišiagala, Vilnius district	14 September 2021
6.	Tyrelis forest, Joniškis distr.	14 September 2021
7.	Šimonys forest, Anykščiai district	14 September 2021
8.	Near Aklasis lake, Jonava district	18 September 2021
9.	Alioniai swamp, Širvintos district	18 September 2021
10.	Near Varninkai educational trail, Trakai district	18 September 2021
11.	Amalva forest, Marijampolė distr.	20 September 2021
12.	Purvai swamp, Biržai district	20 September 2021
13.	Ežerėlis peatbog, Kaunas district	21 September 2021
14.	Žalioji forest, Panevėžys district	21 September 2021
15.	Near Juodlė lake, Kelmė district	21 September 2021
16.	Labanoras forest, Ignalina district	19 September 2021
17.	Snieginis reserve, Švenčionys district	14 September 2021
18.	Dubrava swamp, Kaunas district	22 September 2021
19.	Valkininkai, Varėna district	15 September 2021
20.	Labanoras forest, Molėtai district	12 October 2021
21.	Vaiguva, Kelmė district.	19 October 2021
22.	Čepkeliai reserve, Varėna district	28 September 2021
23.	Kamanai reserve, Akmenė district	14 October 2021
24.	Žuvintas reserve, Alytus district	21 September 2021
25.	Zypliai forest, Šakiai district	19 September 2021

## Data Availability

All data generated during this study are included in this article.

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
