# Peer review of "Determination of Biologically Active Compounds and Antioxidant Capacity In Vitro in Fruit of Small Cranberries (Vaccinium oxycoccos L.) Growing in Natural Habitats in Lithuania"

_antioxidants, 2024, doi:10.3390/antiox13091045_

Round 1

Reviewer 1 Report

In this manuscript, the authors determined the composition of flavonols, proanthocyanidins, anthocyanins, triterpene compounds, 14 and chlorogenic acid in small cranberry fruit samples collected in natural habitats in Lithuania and 15 variations in antioxidant activity of cranberry fruit extracts were determined. The paper is generally well-written and structured, based on comprehensive laboratory research. However, references are the issue because no references were written according to the Reference Style Guide for Antioxidants journal.

Comments:

1) The beginning of the manuscript shows a picture of the geographical area where small cranberry fruit samples were taken. In the results, no attention was paid to the area's climatic conditions while harvesting these fruits. During one month of harvest, was there a rainy or sunny period? How did the climate affect the investigated antioxidant parameters?

2) All images are very poorly visible, that is, visible with a high magnification of the page. Furthermore, symbols related to statistical reliability are not explained below the figures or in the experimental part of the manuscript.

3) Conclusions should be written briefly and clearly, without repeating the results. It is clear that the authors have conducted surveillance on a large number of samples from different areas of Lithuania. However, they should only state their work's most important scientific findings in the conclusions.

Language revisions:

Line 36: grow – grows

Line 46: 352 000 – 352.000

Line 50: have – has

Line 70: protects – protect

Lines 113; 160: was – were

Lines 225; 282; 356: V. oxycoccos – italic font

Lines 290; 342: of the quantitative

Line 406: significantly – significant

Line 414: significantly greater – more significant

Line 436: have an effect - affect

Line 445: of the quality of V. oxycoccos

Line 507: were evaluated

Line 514: the total – a total

Line 529: of quantitative and …

Line 545: in Rekyva

Line 567: the reducing

Lines 571; 576: anti-radical or antiradical?

Author Response

Dear Reviewer,

thank you very much for your valuable time assigned to review our manuscript. Your remarks helped to significantly improve the quality of the manuscript. We tried to take your comments and suggestions into account as much as possible. Please find our responses and explanations in the attached file. 

Yours faithfully,

on behalf of all authors of the manuscript,

Dr. Mindaugas Liaudanskas

Lithuanian University of Health Sciences

Reviewer 2 Report

The article is interesting, but major revisions are suggested to improve its quality.

Title. Exchange antioxidant activity for antioxidant capacity, since an in vitro approach was used.

Abstract. Include a brief conclusion statement.

Introduction. OK

Materials & methods. Include how sampling was carried out, as well as the conditions for transport and storage of the fruits. Provide full details of the chromatographic conditions. Provide the calibration curve data and validation parameters as a supplementary table. 

Results. Authors must avoid repeating the numeric data already observed in tables and figures. Only one hydroxycinnamic acid was detected in cranberry fruits? Or did the authors only had this standard and therefore other phenolic acids were not identified and quantified?

Discussion. The integration and discussion of the antioxidant capacity and the phytochemical profile must be improved.

Conclusions. Authors must avoid including a summary of the results obtained in the study, but properly include a conclusion statement. The limitations of the study must be clearly described.

Author Response

(The authors gave the same response as above.)

Round 2

Reviewer 1 Report

Reviewer's decision

 The authors have carefully considered the reviewer's comments and tried their best to address every one of them. After careful revision, the manuscript has improved toward journal standards. The manuscript is recommended for publication in Antioxidants without any further revisions.

The authors have carefully considered the reviewer's comments and tried their best to address every one of them. After careful revision, the manuscript has improved toward journal standards. The manuscript is recommended for publication in Antioxidants without any further revisions.

Author Response

Dear Reviewer,

thank you very much for your valuable time assigned to review our manuscript. Your remarks helped to significantly improve the quality of the manuscript. We tried to take your comments and suggestions into account as much as possible. All new additions or corrections are highlighted in red for clarity in the manuscript.

We have carefully reviewed the overlaps that are presented in the similarity report. We removed some overlaps from the manuscript. Some overlapping sentences and words have been paraphrased. After reviewing the report, we see that there are many technical overlaps (like “Department of Pharmacognosy, Faculty of Pharmacy, Lithuanian University of Health Sciences”, “Introduction”, “Materials and Methods”, “Plant Material”, and etc.) or simply separate words such as “anthocyanins” and other compounds names which we cannot avoid using. Chromatographic methodologies and their descriptions are our own and their repetition cannot be avoided. We assure you that this manuscript is an original scientific work that has not been published anywhere else, its publication does not violate the rights and interests of other scientists, so we kindly ask you to accept it as suitable for publication in the “Antioxidants” journal.

 We strongly believe that after these corrections our manuscript will be suitable for publication in the “Antioxidants” journal.

Yours faithfully,

on behalf of all authors of the manuscript,

Dr. Mindaugas Liaudanskas

Lithuanian University of Health Sciences

Reviewer 2 Report

The authors included all recommendations suggested to improve the quality of their paper.

The revised manuscript shows a high percentage match according to the iThenticate report. Authors must consider rephrasing several sentences along the manuscript.

Author Response

(The authors gave the same response as above.)
